# Assessment of oxidative stress markers in elderly patients with SARS-CoV-2 infection and potential prognostic implications in the medium and long term

Nestor Vazquez-Agra[1]*, Ana-Teresa Marques-Afonso[1], Anton Cruces-Sande[2]*, Ignacio Novo-Veleiro[1], Antonio Pose-Reino[1], Estefania Mendez-Alvarez[2], Ramon Soto-Otero[2], Alvaro Hermida-Ameijeiras[1]

1 Department of Internal Medicine, University Hospital of Santiago de Compostela, A Coruña, Spain,
2 Laboratory of neurochemistry, Department of Biochemistry and Molecular Biology, Faculty of Medicine, University of Santiago de Compostela, A Coruña, Spain

* Nestor.Vazquez.Agra@sergas.es (NVA); anton.cruces@usc.es (ACS)

## Abstract

We aimed to evaluate the correlation of plasma levels of thiobarbituric acid reactive substances (TBARS) and reduced thiols with morbidity, mortality and immune response during and after SARS-CoV-2 infection. This was an observational study that included inpatients with SARS-CoV-2 infection older than 65 years. The individuals were followed up to the twelfth month post-discharge. Plasma levels of TBARS and reduced thiols were quantified as a measure of lipid and protein oxidation, respectively. Fatal and non-fatal events were evaluated during admission and at the third, sixth and twelfth month post-discharge. Differences in oxidative stress markers between the groups of interest, time to a negative RT-qPCR and time to significant anti-SARS-CoV-2 IgM titers were assessed. We included 61 patients (57% women) with a mean age of 83 years old. After multivariate analysis, we found differences in TBARS and reduced thiol levels between the comparison groups in fatal and non-fatal events during hospital admission. TBARS levels were also correlated with fatal events at the 6th and 12th months post-discharge. One year after hospital discharge, other predictors rather than oxidative stress markers were relevant in the models. The median time to reach significant anti-SARS-CoV-2 IgM titers was lower in patients with low levels of reduced thiols. Assessment of some parameters related to oxidative stress may help identify groups of patients with a higher risk of morbidity, mortality and delayed immune response during and after SARS-CoV-2 infection.

## Introduction

The Severe Acute Respiratory Syndrome Coronavirus 2 (SARS-CoV-2) was identified as the cause of a cluster of pneumonia cases in Wuhan (China) at the end of 2019. Symptoms appeared in six of ten patients and about 15% of individuals had severe disease defined as the presence of dyspnea, hypoxemia and major lung involvement in imaging tests [1, 2].

**Data Availability Statement:** The anonymized database will be available from Open Science Framework (DOI: 10.17605/OSF.IO/R3S8U).

**Funding:** The author(s) received no specific funding for this work.

**Competing interests:** The authors have declared that no competing interests exist.

The literature suggests that an exacerbated inflammatory response in predisposed individuals could be one of the main causes of prognostic differences between patients with SARS-CoV-2 infection [3]. Some studies have pointed to the existence of crosstalk between oxidative stress and inflammation that involves an upregulation of some transcription factors such as Nrf2, NF-KB, and the NLRP3 inflammasome, enhancing pro-inflammatory status [4, 5]. However, non-inflammatory cellular pathways related to redox imbalance and their role in SARS-CoV-2 infection severity and prognosis are not yet fully understood.

Patients with high levels of some inflammatory markers, such as C-reactive protein (CRP), ferritin, fibrinogen and some pro-inflammatory cytokines, such as interleukin 6 (IL-6) and tumor necrosis factor alpha (TNF-α) are at an increased risk of severe complications [6]. However, the role of redox imbalance in the prognosis of SARS-CoV-2 infection in the short and even medium to long term remains poorly understood since most research on SARS-CoV-2 and oxidative stress are reviews of clinical, translational and base studies related to SARS-CoV-2 virulence and pathogenicity or to several diseases that could share some pathogenic mechanisms [7, 8].

The instability and high reactivity of reactive oxygen species (ROS) implies the need for quantification of secondary but more stable organic products derived from their oxidative action. Some of the most quantified oxidative stress markers are thiobarbituric acid reactive substances (TBARS) as a measure of lipid peroxidation and reduced thiols as a measure of protein oxidation. The assessment of TBARS and reduced thiols as a prognostic tool has been widely supported in some diseases and it would not be negligible that some oxidative stress markers could also provide prognostic information in SARS-CoV-2 infection [9–11].

We aimed to quantify plasma TBARS and reduced thiol levels in patients with SARS-CoV-2 infection and to assess whether there is a correlation between these oxidative stress markers and some prognostic variables related to morbidity, mortality and immunity response during and after SARS-CoV-2 infection.

## Material and methods

### Study design and framework

This was an observational study conducted in a SARS-CoV-2 inpatient unit belonging to the Department of Internal Medicine of the University Hospital of Santiago de Compostela (Galicia, Spain) from November/2020 to January/2021. Patients were recruited at hospital admission and followed up during the next 12 months post-discharge. The events of interest were quantified during admission and at the third, sixth and twelfth months post-discharge.

### Participants

We identified patients older than 65 years with confirmed SARS-CoV-2 infection by microbiological criteria who met admission criteria (presence of risk factors or severe disease characterized as the appearance of dyspnea, hypoxemia (O2 saturation lower than 94%) or pulmonary involvement greater than 50%). Patients were randomly selected from the complete cohort of individuals admitted to the SARS-CoV-2 inpatient unit. Patients with a Barthel index of lower than 20 points and those without a record of a laboratory test within the first 7 days since admission were excluded [12].

### Clinical variables

All patients were assessed for demographic characteristics (age and sex), cardiovascular (CV) risk factors (CVRFs) including smoking status (non-smokers versus current or former

smokers), alcohol intake (no consumption versus consumption of any amount), body weight (normal weight versus obesity according to body mass index [BMI]), diabetes mellitus (DM), hyperlipidemia (HLP) and arterial hypertension (AHT). AHT and HLP were defined according to ESC Clinical Practice Guidelines [13, 14]. DM was considered according to the American Diabetes Association guidelines (ADA) [15]. Patients were also investigated for the presence of concomitant chronic cardiovascular, respiratory, and renal diseases [16–18].

The presence of cognitive impairment was confirmed with data collected from the patient's clinical history and in those cases without previous information, we performed the Pfeiffer test [19]. A Barthel index score of lower than or equal to 35 points was considered severe physical dependence [12].

We considered as drug-related variables long-term treatment with renin-angiotensin-aldosterone-system (RAAS) blockers and statins and acute management with antibiotics (azithromycin or others such as B-lactams), systemic steroids and the need for supplemental oxygen therapy.

## Laboratory variables

All patients underwent a blood test between day five and seven of hospital admission. Blood samples were obtained at 08:00 AM following overnight fasting. The collected laboratory parameters were complete blood count, including platelet (PTC) and white blood cell (WBC) count, biochemistry parameters, including serum glucose, creatinine, albumin, triglycerides (TG), ferritin and lactate dehydrogenase, as well as some variables related to coagulation, highlighting fibrinogen levels [20–22].

The diagnosis of SARS-CoV-2 infection was made using upper respiratory tract samples that were collected in compliance with the established protocols [23]. We employed the real-time reverse transcriptase-polymerase chain reaction (RT-qPCR) for SARS-CoV-2 detection and the results were qualitatively reported as negative, positive, or indeterminate. To assess IgM antibodies against the receptor-binding domain (RBD) of the spike protein, we performed enzyme-linked immunosorbent assay (ELISA) and the results were provided quantitatively [24, 25].

## Assessment of oxidative stress markers

Blood samples were collected in tubes with EDTA, centrifuged in less than 1 h after extraction at 1000 G and 4˚C for 10 min. The plasma fraction was deposited at -80˚C for less than 1 month until analysis [26].

The assessment of TBARS is a well-established method for screening and monitoring lipid peroxidation. The most relevant and quantified final products of lipid peroxidation are malondialdehyde (MDA) and 4-hydroxynonenal (4-HNE). Thiobarbituric acid (TBA) forms an adduct with MDA under high temperature (90–100˚C) and acidic conditions yielding a violaceus pigment that can be measured spectrophotometrically at 530–540 nm. The absorbance is directly proportional to the level of plasma lipid peroxides. We followed the protocol of Ohkawa, et al. and TBARS levels were given in micromolar (μmol/L) [27]. According to the literature, physiological concentrations of plasma TBARS range between 0.26 and 3.94 μmol/L [28].

The assessment of reduced thiols is a well-systematized technique for the quantification of protein oxidation. Ellman's technique uses 5,5-dithio-bis-(2-nitrobenzoic acid) as a reagent to form a compound with the sulfhydryl groups of some amino acid residues in proteins, yielding a colorful pigment that can be measured spectrophotometrically at 412 nm and whose absorbance is proportional to reduced thiol levels in plasma proteins. Concentrations were given in

millimolar (mmol/L). According to the literature, physiological levels of plasma reduced thiols range between 0.4 and 0.6 mmol/L [10, 29].

All samples were analyzed in duplicate and a standard calibration for each protocol was performed to obtain a linear model with a coefficient of determination ($R^2$) greater than 98%.

## Outcomes

Fatal events were referred to medical circumstances that led to the patient's death. We grouped them into the follow categories: I) In-hospital fatal events; II) 3rd month post-discharge fatal events; III) 6th month post-discharge fatal events; and IV) 12th month post-discharge fatal events. Non-fatal events were grouped into the following categories: I) In-hospital non-fatal events, which referred to the need for transfer during admission to a critical respiratory care unit; and II) post-discharge non-fatal events, which referred to the need for readmission to hospital for respiratory or cardiovascular disease within the 3rd, 6th, or 12th month post-discharge.

We considered the time during admission to reach a certain threshold of anti-SARS-CoV-2 IgM titers as a possible indicator of immune response efficiency and the selected cut-off was the median anti-SARS-CoV-2 IgM titers of the sample. The time during admission for a negative RT-qPCR was considered a measure of virus clearance efficiency.

## Calculation of sample size and treatment of variables

For an unknown population size, considering a variance of 0.5 and 0.05 with a threshold for the mean difference to be detected of 0.5 and 0.05 units for TBARS and reduced thiols, respectively, the sample size calculated to estimate mean differences between the groups with a 95% confidence interval was a minimum of n = 30 patients [30]. Variables were collected according to data provided by the regional digital health records (IANUS) belonging to the Galician (Spain) Health Service (SERGAS). Most of the clinical variables were coded as qualitative ones and laboratory parameters, including TBARS and reduce thiols, were collected as continuous quantitative variables. Outcomes were coded as dichotomous qualitative variables.

## Ethical approval

The Research Ethics Committee of Santiago de Compostela-Lugo approved this study (reference number 2020/578). All procedures were in accordance with the ethical standards of the responsible committee on human experimentation and with the Helsinki Declaration of 1975. Written informed consent was obtained from all patients for being included in the study.

## Statistical analysis

Statistical analysis was performed using SPSS 22.0 statistical software (SPSS Inc, Chicago, IL). First, we performed a descriptive analysis in which the frequencies of qualitative variables were expressed as number (n) and percentage (%). The Kolmogorov-Smirnov test was used to determine whether quantitative continuous variables were normally distributed. For those variables with a non-normal distribution, we performed a logarithmic transformation. Normally distributed variables were expressed as mean (m) and standard deviation (± SD), normally distributed variables after transformation were back-transformed and expressed as mean with 95% confidence interval (95%CI) and non-normally distributed ones were expressed as median and interquartile range (IQR). A missing value analysis was carried out on those variables with more than 5% missing values. We performed a comparative analysis between the groups of patients according to the outcome variables. The chi-square test was used to

compare categorical variables, while quantitative variables were compared using the Student's t-test or Mann–Whitney U test as appropriate.

We performed a binary logistic regression analysis for fatal and non-fatal events during admission and at the $3^{rd}$, $6^{th}$ and $12^{th}$ months post-discharge using a non-automatic analysis procedure. Variables that showed clinical or statistical relevant differences (P-value < 0.1) in the univariate analysis were included. The validity of the model was evaluated using the Omnibus test in which a P-value of lower than 0.05 was considered necessary to assume that the current model was better than the null model. We provided a model summary with the deviance (-2 log-likelihood ratio test (-2LL)), coefficient of determination ($R^2$) and the overall accuracy score. The parameters of those variables in the model were the non-standardized Beta coefficients (B), P-value of the Wald test and 95%CI for the coefficients. The variables with a P-value of lower than 0.1 (P< 0.150) were kept in the model. A P-value of lower than 0.05 (P< 0.05) was considered for statistical significance.

We developed a survival analysis model using the Kaplan–Meier estimator for time to reach negative RT-qPCR and time to achieve significant anti-SARS-CoV-2 IgM titers during admission based on plasma levels of TBARS and reduced thiols and using as cut-off points the physiological limits of TBARS (4.0 μmol/L) and reduced thiol (0.40 mmol/L) levels. We compared the evolution of the variables of interest as a function of time between patients with lower or equal levels and higher levels of the oxidative stress markers. The relevance of the results was evaluated using the Log-Rank test and a P-value of lower than 0.05 (P< 0.05) was considered for statistical significance.

## Results

We included 61 patients (57% women) with a mean age of 83 years, and among them, 42 (69%), 31 (51%) and 15 (25%) had AHT, HLP and DM, respectively. A total of 27 (44%) suffered from chronic heart failure (HF). Approximately one of four individuals had tobacco or alcohol abuse. The mean levels of plasma TBARS and reduced thiols were 2.91 (2.06–4.13) μmol/L and 0.47 (0.40–0.55) mmol/L respectively. The median anti-SARS-CoV-2 IgM titers were 27 (40) U/mL. All the results are shown in Table 1.

### Fatal events

Results of the univariate analysis are summarized in Table 1. We did not find differences in age or sex between the comparison groups except at the $6^{th}$ month post-discharge in which patients with fatal events were younger (P = 0.016). Tobacco abuse was more frequent in patients with fatal events after the $3^{rd}$ month post-discharge. We found a higher frequency of AHT in the worst prognosis groups bordering statistical significance at the $3^{rd}$ month post-discharge (P = 0.088). The presence of obesity was generally lower in patients with a fatal event with relevant results at the $3^{rd}$ (P = 0.038), $6^{th}$ (P = 0.011) and $12^{th}$ (P = 0.011) months post-discharge. Except during admission, the presence of HF was higher in the groups with the worst prognosis.

For all groups, there was a higher frequency of azithromycin and systemic steroids use reaching statistical significance at the $12^{th}$ month post-discharge (P = 0.009 and P = 0.026, respectively). Patients with a fatal event had higher levels of FPG than the comparison groups and individuals with a worse prognosis tended to have higher TG, LDH and ferritin levels. We saw a tendency for lower PTC and lymphocyte count with higher levels of neutrophils in patients with fatal events.

Patients who had a fatal event showed higher TBARS levels than controls measured in μmol/L and expressed as mean and 95%CI. These differences reached relevant results in the

**Table 1. Clinical and laboratory features of the comparison groups attending to mortality.**

| Variables | Total sample | In-hospital fatal events | | Post-discharge fatal events | | | | | |
| --- | --- | --- | --- | --- | --- | --- | --- | --- | --- |
| | | | | 3rd month | | 6th month | | 12th month | |
| | n = 61 | No n = 57 | Yes n = 4 | No n = 54 | Yes n = 7 | No n = 52 | Yes n = 9 | No n = 47 | Yes n = 13 |
| Age (years)† | 83 ± 7 | 83 ± 8 | 81 ± 5 | 83 ± 8 | 80 ± 4 | 83 ± 7 | 78 ± 4** | 83 ± 7 | 81 ± 6 |
| Sex (women)‡ | 35 (57) | 33 (58) | 2 (50) | 31 (57) | 4 (57) | 30 (58) | 5 (56) | 28 (60) | 6 (46) |
| Obesity‡ | 23 (38) | 23 (40) | 0 (0) | 23 (43) | 0 (0)** | 23 (44) | 0 (0)** | 22 (47) | 1 (8)** |
| Alcohol intake‡ | 15 (25) | 14 (25) | 1 (25) | 14 (26) | 1 (14) | 13 (25) | 2 (22) | 12 (26) | 3 (23) |
| Current/Former smokers‡ | 16 (26) | 15 (26) | 1 (25) | 14 (26) | 2 (29) | 13 (25) | 3 (33) | 10 (21) | 6 (46)* |
| AHT‡ | 42 (69) | 38 (67) | 4 (100) | 35 (65) | 7 (100)* | 34 (65) | 8 (89) | 30 (64) | 11 (85) |
| HLP‡ | 31 (51) | 30 (53) | 1 (25) | 28 (52) | 3 (43) | 27 (52) | 4 (44) | 24 (51) | 7 (54) |
| DM‡ | 15 (25) | 15 (26) | 0 (0) | 15 (28) | 0 (0) | 13 (25) | 2 (22) | 13 (28) | 2 (15) |
| HF‡ | 27 (44) | 26 (46) | 1 (25) | 23 (43) | 4 (57) | 22 (42) | 5 (56) | 18 (38) | 8 (62) |
| Cognitive impairment‡ | 28 (46) | 25 (44) | 3 (75) | 24 (44) | 4 (57) | 23 (44) | 5 (56) | 22 (47) | 6 (46) |
| Barthel index (≤ 35)‡ | 9 (15) | 8 (14) | 1 (25) | 8 (15) | 1 (14) | 8 (15) | 1 (11) | 7 (15) | 2 (15) |
| RAAS blockers‡ | 22 (36) | 22 (39) | 0 (0) | 20 (37) | 2 (29) | 20 (39) | 2 (22) | 19 (40) | 3 (23) |
| Statins‡ | 22 (36) | 21 (37) | 1 (25) | 20 (37) | 2 (29) | 19 (37) | 3 (33) | 16 (34) | 6 (46) |
| Azithromycin‡ | 22 (36) | 19 (33) | 3 (75) | 17 (32) | 5 (71) | 17 (33) | 5 (56) | 13 (28) | 9 (69)** |
| Systemic steroids‡ | 45 (74) | 41 (72) | 4 (100) | 38 (70) | 7 (100) | 36 (69) | 9 (100)* | 32 (68) | 13 (100)** |
| Oxygen therapy‡ | 23 (38) | 20 (35) | 3 (75) | 19 (35) | 4 (57) | 18 (35) | 5 (56) | 17 (36) | 6 (46) |
| FPG (mg/dL)�ⱺ | 95 (35) | 95 (33) | 127 (249) | 95 (30) | 119(43) | 95 (27) | 119 (46) | 95 (29) | 117 (44) |
| Creatinine (mg/dL)†† | 0.9 (0.6–1.3) | 0.9 (0.6–1.3) | 0.7 (0.6–0.8) | 0.9 (0.6–1.3) | 0.9 (0.6–1.3) | 0.9 (0.6–1.3) | 0.9 (0.7–1.2) | 0.9 (0.6–1.3) | 0.9 (0.7–1.2) |
| Albumin (g/dL)†† | 3.8 (3.4–4.2) | 3.8 (3.3–4.3) | 3.6 (3.4–3.9) | 3.8 (3.4–4.2) | 3.7 (3.4–4.0) | 3.8 (3.4–4.3) | 3.7 (3.3–4.1) | 3.9 (3.5–4.3) | 3.5 (3.1–4.0)** |
| TG (mg/dL)†† | 109 (64–186) | 107 (63–184) | 138 (90–212) | 109 (64–185) | 118 (66–209) | 108 (63–185) | 119 (73–199) | 108 (62–186) | 120 (73–195) |
| LDH (IU/L)†† | 401 (286–563) | 393 (281–550) | 545 (426–697)* | 394 (281–554) | 466 (341–637) | 396 (281–559) | 434 (317–593) | 3901 (282–542) | 454 (313–660) |
| Ferritin (ng/mL)†† | 200 (60–666) | 185 (56–613) | 605 (323–1132)* | 186 (55–634) | 345 (140–855) | 180 (55–590) | 374 (120–1166)* | 176 (53–590) | 313 (98–996) |
| Fibrinogen (mg/dL)�ⱺ | 472 (222) | 466 (224) | 495 (222) | 472 (217) | 391 (309) | 472 (217) | 423 (268) | 472 (217) | 438 (199) |
| PTC (10³/μL)†† | 189 (127–281) | 192 (129–284) | 160 (103–251) | 190 (128–284) | 183 (123–271) | 192 (128–287) | 178 (124–255) | 190 (127–283) | 175 (128–241) |
| Neutrophils (10³/μL)ⱺ | 3.4 (4.1) | 3.3 (3.6) | 8.7 (7.6) | 3.2 (3.6) | 7.1 (6.1) | 3.2 (3.4) | 7.1 (6.3) | 3.1 (3.2) | 6.4 (5.0) |
| Lymphocytes (10³/μL)†† | 1.2 (0.7–2.3) | 1.3 (0.7–2.4) | 0.7 (0.4–1.2)* | 1.3 (0.7–2.4) | 0.9 (0.5–1.7) | 1.3 (0.7–2.4) | 0.9 (0.5–1.7) | 1.3 (0.7–2.5) | 1.0 (0.6–1.6)* |

AHT–Arterial hypertension. HLP–Hyperlipidemia. DM–Diabetes mellitus. HF–Heart failure. RAAS–Renin-angiotensin-aldosterone-system. FPG–Fasting plasma glucose. TG–Triglycerides. LDH–Lactate dehydrogenase. PTC–Platelet count. Results expressed as † refer to mean ± standard deviation. Results expressed as ‡ refer to number (percentage). Results expressed as †† refer to mean (95%CI) after back-transformation. Results expressed as ⱺ refer to median (Interquartile range).

* Indicated comparison with patients without in-hospital, 3rd, 6th, or 12th month post-discharge fatal events ($P < 0.10$).

** Indicated comparison with patients without in-hospital, 3rd, 6th, or 12th month post-discharge fatal events ($P < 0.05$).

following groups: In-hospital (no: 2.84 (2.03–3.97), yes: 4.20 (2.93–6.01); $P = 0.029$), 3rd (no: 2.83 (2.00–3.98), yes: 3.69 (2.71–5.04); $P = 0.054$) and 6th (no: 2.81 (2.02–3.92), yes: 3.57 (2.42–5.26); $P = 0.058$) months post-discharge fatal events. The levels of reduced thiols measured in mmol/L and expressed as mean and 95%CI were lower in patients who suffered a fatal event. Such differences reached relevant results in the following groups: 3rd (no: 0.47 (0.40–0.56), yes: 0.42 (0.37–0.46); $P = 0.059$), 6th (no: 0.48 (0.40–0.56), yes: 0.41 (0.36–0.48); $P = 0.026$) and 12th (no: 0.48 (0.41–0.56), yes: 0.43 (0.36–0.51); $P = 0.034$) month post-discharge fatal events. The results are shown in Fig 1A and 1B.

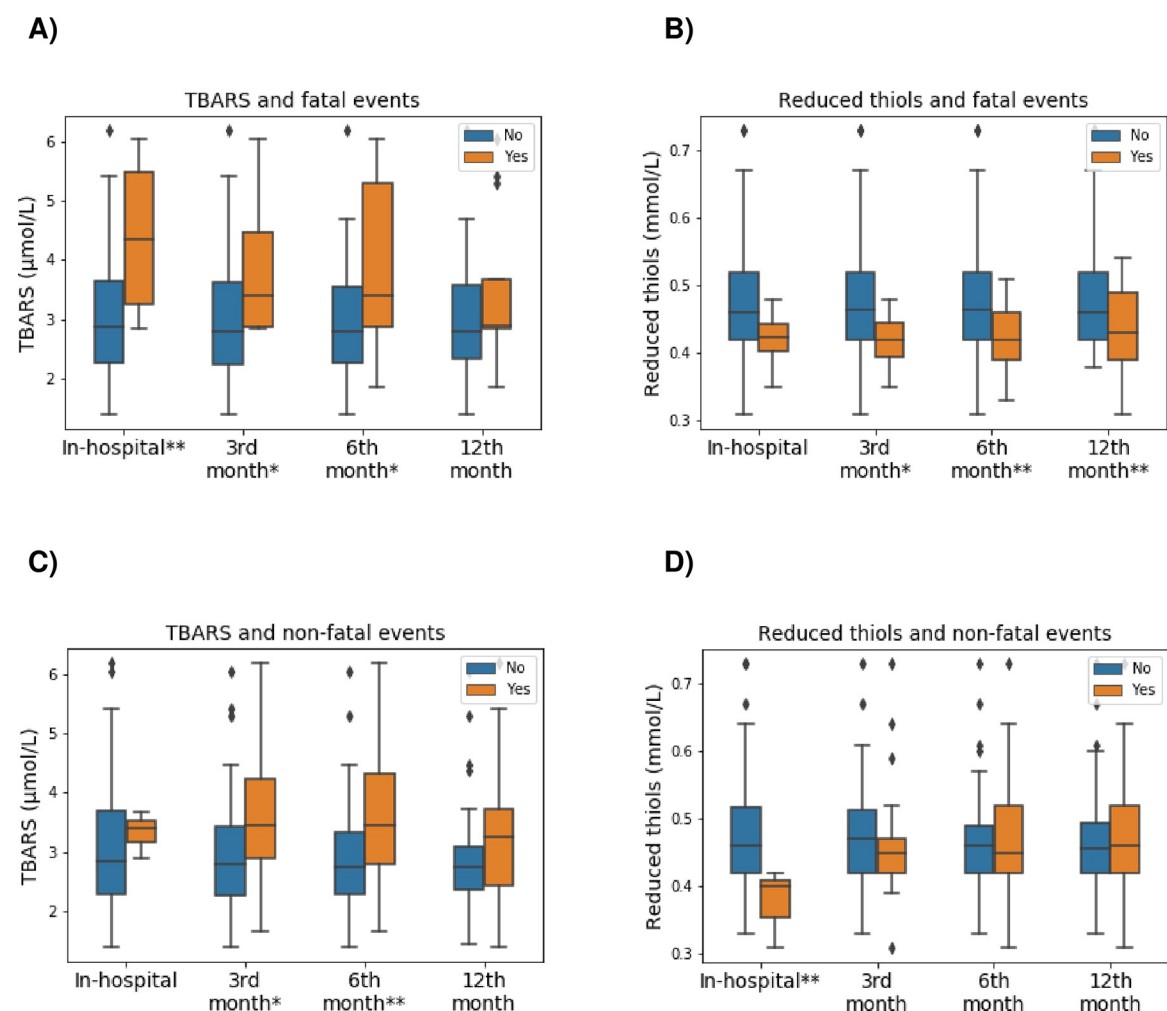

**Fig 1. Differences between plasma levels of TBARS and reduced thiols in the groups of interest.** (A) TBARS and fatal events.(B) Reduced thiols and fatal events. (C) TBARS and non-fatal events. (D) Reduced thiols and non-fatal events. The results are shown as mean ± standard deviation. TBARS–Tiobarbituric acid reactive substances. * Refers to *P*-value < 0.10. ** Refers to *P*-value < 0.05.

### Non-fatal events

The results of the comparison groups are summarized in Table 2. We did not find differences in age or sex between the comparison groups. Tobacco abuse was generally more frequent in patients with a non-fatal event and we found a higher percentage of AHT in the poor prognosis groups bordering on statistical significance at the 3[rd] ($P$ = 0.064) month post-discharge. The presence of HF was higher in the worst prognosis groups.

For all groups, there was a higher frequency of acute use of azithromycin that reached statistical significance during hospital admission ($P$ = 0.043) and the frequency of systemic steroid use was also higher in patients with a non-fatal event. We found clinical differences in TG levels between the comparison groups during admission. LDH levels and PTC were higher in patients with non-fatal events, reaching statistical significance at the 6[th] ($P$ = 0.044) and 12[th] (0.039) month post-discharge respectively. Except for the 12[th] month post-discharge, higher ferritin levels showed a clinical correlation with non-fatal events.

Patients who had a non-fatal event presented higher TBARS levels than controls measured in µmol/L and expressed as mean and 95%CI. Such differences reached relevant results in the

**Table 2. Clinical and laboratory features of the comparison groups attending to non-fatal events.**

| Variables | In-hospital non-fatal events | | Post-discharge non-fatal events | | | | | |
|---|---|---|---|---|---|---|---|---|
| | | | 3rd month | | 6th month | | 12th month | |
| | No | Yes | No | Yes | No | Yes | No | Yes |
| | n = 58 | n = 3 | n = 44 | n = 17 | n = 38 | n = 23 | n = 28 | n = 33 |
| Age (years)† | 82 ± 7 | 84 ± 6 | 83 ± 8 | 83 ± 5 | 83 ± 8 | 82 ± 6 | 82 ± 7 | 83 ± 8 |
| Gender (women)‡ | 33 (57) | 2 (67) | 26 (59) | 9 (53) | 23 (61) | 12 (52) | 18 (64) | 17 (52) |
| Obesity‡ | 23 (40) | 0 (0) | 17 (39) | 6 (35) | 15 (40) | 8 (35) | 10 (36) | 13 (39) |
| Alcohol intake‡ | 15 (26) | 0 (0) | 10 (23) | 5 (29) | 8 (21) | 7 (30) | 5 (18) | 10 (30) |
| Current/Former smokers‡ | 15 (26) | 1 (33) | 10 (23) | 6 (35) | 7 (18) | 9 (39) | 5 (18) | 11 (33) |
| AHT‡ | 39 (67) | 3 (100) | 27 (61) | 15 (88)* | 24 (63) | 18 (78) | 18 (64) | 24 (73) |
| HLP‡ | 29 (50) | 2 (67) | 21 (48) | 10 (59) | 18 (47) | 13 (57) | 13 (46) | 18 (55) |
| DM‡ | 15 (26) | 0 (0) | 10 (23) | 5 (29) | 8 (21) | 7 (30) | 6 (21) | 9 (27) |
| HF‡ | 25 (43) | 2 (67) | 17 (39) | 10 (59) | 14 (37) | 13 (57) | 9 (32) | 18 (55) |
| Cognitive impairment‡ | 27 (47) | 1 (33) | 21 (48) | 7 (41) | 19 (50) | 9 (39) | 14 (50) | 14 (42) |
| Barthel index (≤ 35)‡ | 9 (16) | 0 (0) | 8 (18) | 1 (6) | 7 (18) | 2 (9) | 7 (25) | 2 (6)* |
| RAAS blockers‡ | 20 (35) | 2 (67) | 16 (36) | 6 (35) | 16 (42) | 6 (26) | 10 (36) | 12 (36) |
| Statins‡ | 20 (35) | 2 (67) | 15 (34) | 7 (41) | 12 (32) | 10 (44) | 9 (32) | 13 (39) |
| Azithromycin‡ | 19 (33) | 3 (100)** | 14 (32) | 8 (47) | 11 (29) | 11 (48) | 8 (29) | 14 (42) |
| Systemic steroids‡ | 42 (72) | 3 (100) | 31 (71) | 14 (82) | 26 (68) | 19 (83) | 20 (71) | 25 (76) |
| Oxygen therapy‡ | 21 (36) | 2 (67) | 15 (34) | 8 (47) | 14 (37) | 9 (39) | 9 (32) | 14 (42) |
| FPG (mg/dL)¶ | 95 (35) | 117 (-) | 94 (25) | 115 (57)* | 94 (26) | 106 (45) | 95 (28) | 96 (43) |
| Creatinine (mg/dL)†† | 0.9 (0.6–1.3) | 0.8 (0.7–1.0) | 0.9 (0.6–1.3) | 0.9 (0.6–1.4) | 0.9 (0.7–1.3) | 0.9 (0.6–1.3) | 0.9 (0.6–1.2) | 0.9 (0.7–1.3) |
| Albumin (g/dL)†† | 3.8 (3.4–4.2) | 3.4 (3.0–3.9) | 3.8 (3.4–4.3) | 3.7 (3.4–4.1) | 3.8 (3.4–4.3) | 3.7 (3.4–4.2) | 3.8 (3.3–4.3) | 3.8 (3.4–4.2) |
| TG (mg/dL)†† | 107 (63–182) | 179 (144–222) | 120 (64–188) | 110 (66–185) | 110 (63–193) | 110 (68–178) | 117 (82–167) | 104 (55–197) |
| LDH (IU/L)†† | 397 (282–559) | 506 (423–605) | 387 (283–530) | 443 (300–653) | 376 (281–502) | 449 (305–661)** | 385 (291–510) | 417 (285–610) |
| Ferritin (ng/mL)†† | 193 (57–653) | 387 (194–775) | 179 (53–607) | 266 (85–834) | 186 (54–648) | 225 (72–705) | 239 (66–869) | 172 (56–528) |
| Fibrinogen (mg/dL)¶ | 472 (216) | 374 (–) | 465 (228) | 466 (217) | 473 (227) | 449 (219) | 465 (222) | 466 (224) |
| PTC (10³/μL)†† | 189 (126–282) | 205 (147–285) | 181 (128–257) | 213 (131–346) | 179 (124–259) | 208 (135–320) | 169 (119–241) | 209 (139–319)** |
| Neutrophils (10³/μL)¶ | 3.2 (4.1) | 6.3 (–) | 3.2 (3.4) | 4.7 (5.4) | 3.1 (4.0) | 4.4 (4.6) | 3.1 (4.6) | 3.7 (4.3) |
| Lymphocytes(10³/μL) †† | 1.3 (0.7–2.4) | 0.7 (0.6–1.0) | 1.3 (0.8–2.2) | 1.1 (0.5–2.6) | 1.3 (0.8–2.3) | 1.1 (0.5–2.3) | 1.4 (0.8–2.3) | 1.2 (0.6–2.3) |

AHT–Arterial hypertension. HLP–Hyperlipidemia. DM–Diabetes mellitus. HF–Heart failure. RAAS–Renin-angiotensin-aldosterone-system. FPG–Fasting plasma glucose. TG–Triglycerides. LDH–Lactate dehydrogenase. PTC–Platelet count. Results expressed as † refer to mean ± standard deviation. Results expressed as ‡ refer to number (percentage). Results expressed as †† refer to mean (95%CI) after back-transformation. Results expressed as ¶ refer to median (Interquartile range).

* Indicated comparison with patients without in-hospital, 3rd, 6th, or 12th month post-discharge non-fatal events ($P < 0.10$).

** Indicated comparison with patients without in-hospital, 3rd, 6th, or 12th month post-discharge fatal events ($P < 0.05$).

following groups: 3rd (no: 2.89 (2.03–4.13), yes: 3.31 (2.93–3.73); $P = 0.074$) and 6th (no: 2.77 (1.96–3.91), yes: 3.31 (2.37–4.62); $P = 0.037$) months post-discharge non-fatal events. The levels of reduced thiols measured in mmol/L and expressed as mean and 95%CI were lower in patients who suffered non-fatal events. Such differences reached relevant results in the following groups: In-hospital (no: 0.47 (0.40–0.56), yes: 0.37 (0.32–0.43); $P = 0.019$) non-fatal events. The results are extended in Fig 1C and 1D.

## Multivariate analysis

We performed a multivariate analysis for fatal and non-fatal events during admission and at the 3rd, 6th and 12th months post-discharge using binary logistic regression. We considered some clinical and laboratory variables that could influence TBARS and reduced thiol levels or

**Table 3. Multiple linear regression models for fatal and non-fatal events.**

| Variables | B | *P*-value | Odds ratio | 95%CI | |
|---|---|---|---|---|---|
| | | | | Inferior | Superior |
| ***In-hospital*** | | | | | |
| **Fatal events.** *P-value (Omnibus test) = 0.006, -2LL = 17.206, R² (Nagelkerke) = 0.477, Total accuracy: 96.7%.* | | | | | |
| TBARS (µmol/L) | 1.489 | 0.023 | 4.433 | 1.228 | 16.004 |
| Neutrophils (10³/µL) | 0.463 | 0.035 | 1.589 | 1.033 | 2.445 |
| PTC (10³/µL) | -0.017 | 0.114 | 0.983 | 0.963 | 1.004 |
| **Non-fatal events.** *P-value (Omnibus test) = 0.010, -2LL = 14.674, R² (Nagelkerke) = 0.434, Total accuracy: 95.1%.* | | | | | |
| Reduced Thiols (mmol/L) | -35.460 | 0.033 | 0.001 | 0.000 | 0.055 |
| Fibrinogen (mg/dL) | -0.006 | 0.157 | 0.994 | 0.985 | 1.002 |
| ***3rd month*** | | | | | |
| **Fatal events.** *P-value (Omnibus test) = 0.011, -2LL = 32.337, R² (Nagelkerke) = 0.327, Total accuracy: 86.9%.* | | | | | |
| Azitromicine (yes) | 2.324 | 0.038 | 10.215 | 1.133 | 92.121 |
| TBARS (µmol/L) | 0.946 | 0.030 | 2.575 | 1.097 | 6.044 |
| Fibrinogen (mg/dL) | -0.007 | 0.080 | 0.993 | 0.985 | 1.001 |
| **Non-fatal events.** *P-value (Omnibus test) = 0.021, -2LL = 64.466, R² (Nagelkerke) = 0.171, Total accuracy: 72.1%.* | | | | | |
| AHT (yes) | 1.628 | 0.053 | 5.094 | 0.978 | 26.539 |
| TBARS (µmol/L) | 0.478 | 0.085 | 1.613 | 0.937 | 2.778 |
| ***6th month*** | | | | | |
| **Fatal events.** *P-value (Omnibus test) = 0.013, -2LL = 40.312, R² (Nagelkerke) = 0.285, Total accuracy: 83.6%.* | | | | | |
| Ferritin (ng/mL) | 0.002 | 0.032 | 1.002 | 1.000 | 1.003 |
| TBARS (µmol/L) | 0.746 | 0.028 | 1.886 | 1.003 | 3.545 |
| Fibrinogen (mg/dL) | -0.006 | 0.078 | 0.994 | 0.988 | 1.001 |
| **Non-fatal events.** *P-value (Omnibus test) = 0.018, -2LL = 72.853, R² (Nagelkerke) = 0.167, Total accuracy: 68.9%.* | | | | | |
| LDH (IU/L) | 0.003 | 0.065 | 1.003 | 1.000 | 1.007 |
| TBARS (µmol/L) | 0.472 | 0.076 | 1.603 | 0.951 | 2.703 |
| ***12th month*** | | | | | |
| **Fatal events.** *P-value (Omnibus test)< 0.001, -2LL = 43.762, R² (Nagelkerke) = 0.418, Total accuracy: 85%.* | | | | | |
| Obesity (yes) | -2.969 | 0.018 | 0.051 | 0.004 | 0.605 |
| Tobacco abuse (yes) | 1.949 | 0.030 | 7.022 | 1.212 | 40.671 |
| Azitromicine use (yes) | 1.747 | 0.026 | 5.735 | 1.230 | 26.741 |
| **Non-fatal events.** *P-value (Omnibus test) = 0.002, -2LL = 65.011, R² (Nagelkerke) = 0.360, Total accuracy: 67.2%.* | | | | | |
| HF (yes) | 1266 | 0.057 | 3.548 | 0.962 | 13.083 |
| Barthel index (≤ 35) | -1.689 | 0.067 | 0.185 | 0.030 | 1.129 |
| PTC (10³/µL) | 0.011 | 0.020 | 1.011 | 1.002 | 1.021 |
| TBARS (µmol/L) | 0.572 | 0.086 | 1.772 | 0.922 | 3.405 |
| Ferritin (ng/mL) | -0.002 | 0.104 | 0.998 | 0.997 | 1.000 |

TBARS—Thiobarbituric acid reactive substances. PTC—Platelet count. AHT—Arterial hypertension. LDH—Lactate dehydrogenase. HF—Heart failure. -2LL—-2 Log-Likelihood.

be relevant to the outcomes as follows: age, sex, obesity, toxic habits (tobacco abuse and alcohol intake), CVR variables (AHT, DM and HLP), HF, Barthel index, chronic treatments (statins and RAAS blockers), acute treatments (Azithromycin and systemic steroids) and levels of some analytical parameters (FPG, creatinine, albumin, ferritin, fibrinogen and LDH). Table 3 shows the model summaries and the parameters of those variables that were kept in the model for fatal and non-fatal events during admission and at the 3rd, 6th and 12th months post-discharge.

TBARS levels were correlated with mortality during admission (B = 1.489, P = 0.023, OR = 4.433, 95%CI: 1.228–16.004) and at the 3rd (B = 0.946, P = 0.030, OR = 2.575, 95%CI: 1.097–6.044) and 6th (B = 0.746, P = 0.028, OR = 1.886, 95%CI: 1.003–3.545) month post-discharge. The results for reduced thiols did not reach statistical significance.

Non-fatal events were strongly correlated with reduced thiol levels only during admission (B = -35.460, P = 0.033, OR = 0.001, 95%CI: 0.000–0.055). We found a clinical correlation between TBARS levels and non-fatal events at the 3rd and 6th months post-discharge.

One year after hospital discharge, the variables within the model for fatal events were the presence of obesity, tobacco abuse and use of azithromycin during hospital admission. Non-fatal events were correlated with the presence of HF, Barthel index of equal to or lower than 35 points, PTC, TBARS and ferritin levels during hospital admission.

## Correlation between oxidative stress markers and time to a negative RT-qPCR

Fig 2 represents a survival analysis that shows on the ordinate axis the percentage of patients with a positive RT-qPCR for SARS-CoV-2 and on the abscissa axis the time to evolution in days since the hospital admission. At the beginning, the percentage of patients with a positive RT-qPCR for SARS-CoV-2 was 100% and progressively decreased with the evolution of the disease. The black curve in Fig 2A refers to patients with reduced thiol levels equal to or less than 0.40 mmol/L, while the gray curve refers to individuals with reduced thiol levels higher than 0.40 mmol/L. The black curve in Fig 2B refers to patients with TBARS levels higher than 4.0 μmol/L, while the gray curve refers to individuals with TBARS levels equal to or less than 4.0 μmol/L. The Log-Rank test showed that there were no differences in the median of days to achieving a negative RT-qPCR for SARS-CoV-2 based on different levels of TBARS or reduced thiols.

## Correlation between oxidative stress markers and time to significant anti-SARS-CoV-2 IgM titers

Fig 3 represents a survival analysis that shows on the ordinate axis the percentage of patients with anti-SARS-CoV-2 IgM levels below the median (27 U/mL) and on the abscissa axis the time of evolution in days since the hospital admission. At the beginning, the percentage of patients with non-significant titers was 100% and progressively decreased with the evolution of the disease. The black curve in Fig 3A refers to patients with reduced thiol levels equal to or less than 0.40 mmol/L, while the gray curve refers to individuals with reduced thiol levels higher than 0.40 mmol/L. The black curves in Fig 3B refer to patients with TBARS levels higher than 4.0 μmol/L, while the gray curves refer to individuals with TBARS levels equal to or less than 4.0 μmol/L. We found that the median time to achieving anti-SARS-CoV-2 IgM titers of higher than 27 U/mL was 22 and 14 days for patients with reduced thiol levels greater than 0.40 mmol/L and lower than or equal to 0.40 mmol/L, respectively (P-value = 0.006).

## Discussion

The results are summarized as follows: 1) We found differences in the levels of oxidative stress markers for fatal and non-fatal events during admission and at the 3rd, 6th and 12th month post-discharge that reached clinical relevance. 2) Differences in TBARS and reduced thiol levels showed a correlation with the presence of fatal and non-fatal events during admission, respectively. 3) We found a correlation between TBARS levels and fatal events at the 3rd and 6th month post-discharge. 4) One year after hospital discharge, other predictors more related

**A)**

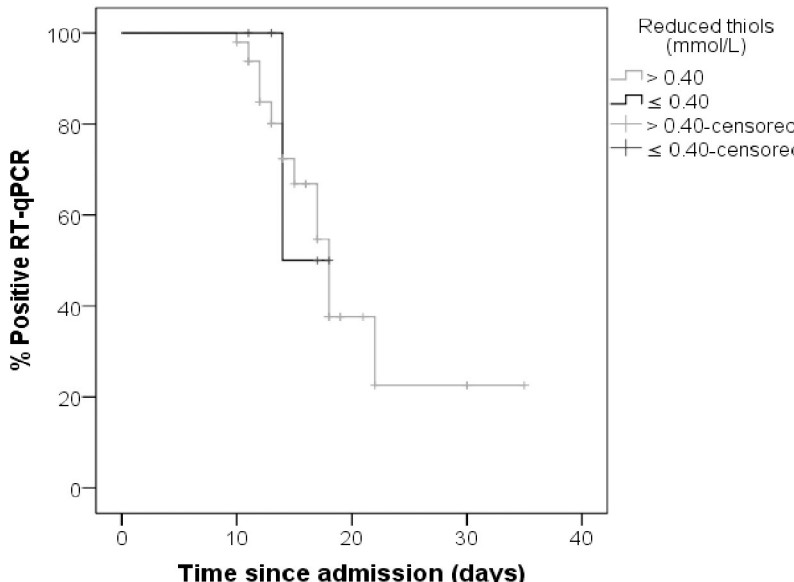

**B)**

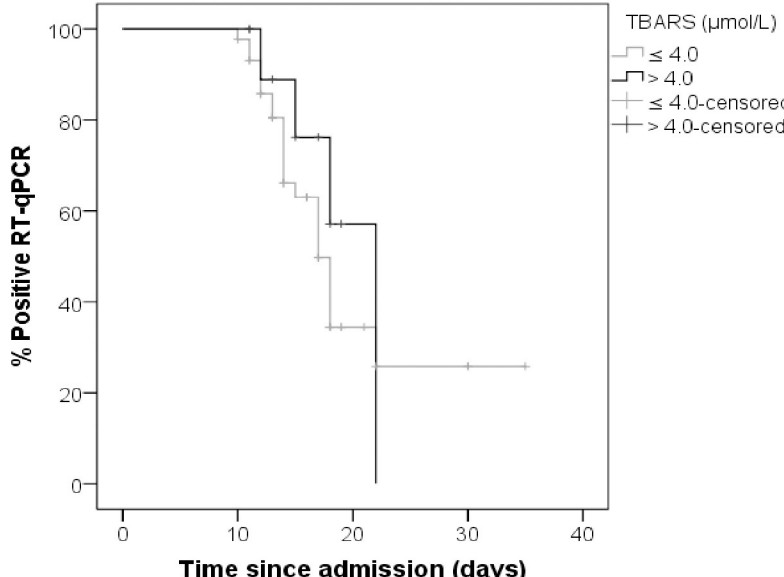

**Fig 2. Survival analysis.** Correlation between oxidative stress markers and time to a negative RT-qPCR. (A) Correlation of plasma levels of reduced thiols with time to a negative RT-qPCR. Log Rank (Mantel-Cox), *P*-value = 0.802; (B) Correlation of plasma levels of TBARS with time to a negative RT-qPCR. Log Rank (Mantel-Cox), *P*-value = 0.396. TBARS–Tiobarbituric acid reactive substances. % Positive RT-qPCR–Percentage of patients with a positive Real-time reverse transcriptase-polymerase chain reaction for SARS-CoV-2.

**A)**

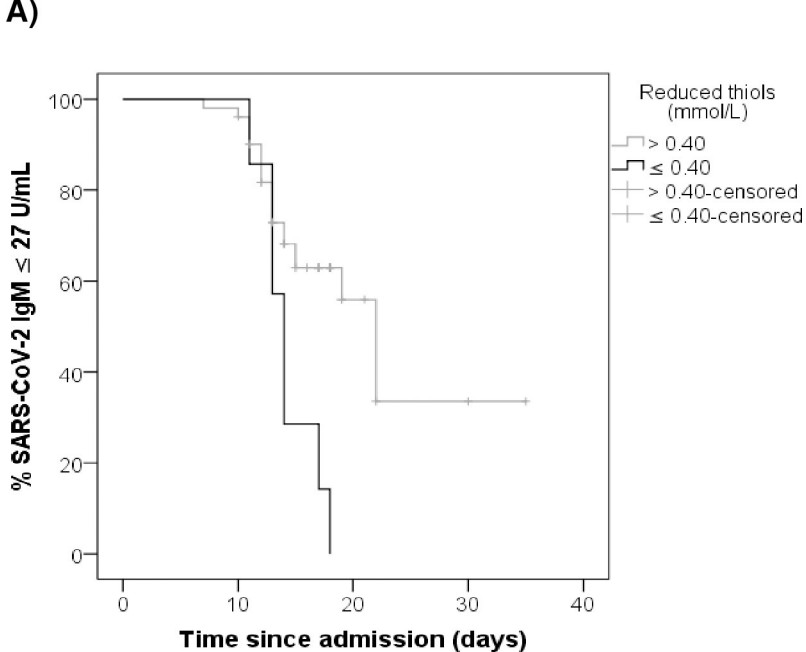

**B)**

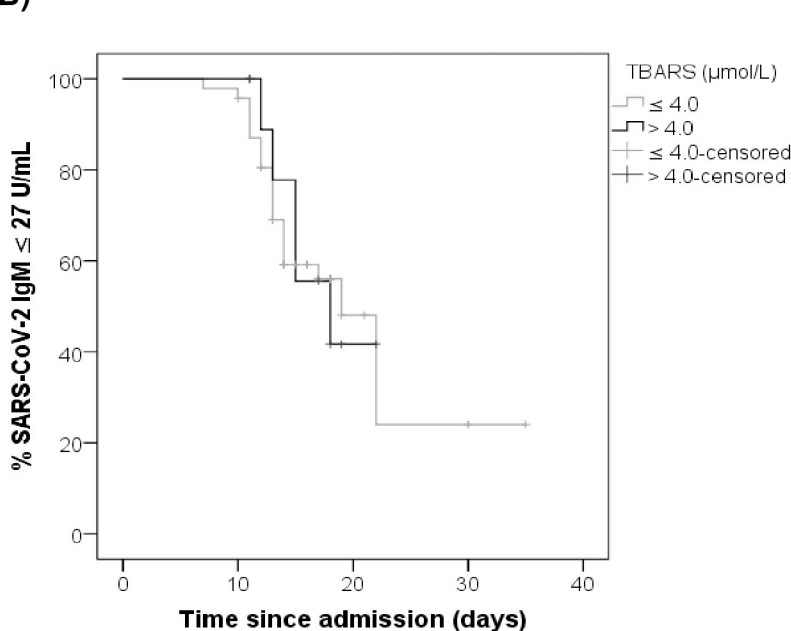

**Fig 3. Correlation between oxidative stress markers and time to significant anti-SARS-CoV-2 IgM titers.** (A) Correlation of plasma levels of reduced thiols with time to significant anti-SARS-CoV-2 IgM titers. Log Rank (Mantel-Cox), *P*-value = 0.006. (B) Correlation of plasma levels of TBARS with time to significant anti-SARS-CoV-2 IgM titers. Log Rank (Mantel-Cox), *P*-value = 0.841. TBARS–Tiobarbituric acid reactive substances. % SARS-CoV-2 IgM ≤ 27 U/mL–Percentage of patients with anti-SARS-CoV-2 IgM titers equal to or lower than 27 U/mL.

to concurrent cardiovascular risk factors and chronic diseases rather than oxidative stress markers were relevant in the models. 5) The median time to reach significant anti-SARS-CoV-2 IgM titers was lower in patients with low levels of reduced thiols.

The role of multiple clinical and laboratory markers as indicators of severity in SARS-CoV-2 infection has been widely reported [31]. Oxidative stress is a global process in which all organic molecules are involved and lipids, glucids, proteins and nucleic acids are the main acceptors of unpaired electrons from unstable ROS [32]. Several studies have established the possible role of some oxidative stress markers in the short-term prognosis of SARS-CoV-2 infection and we also found that the presence of abnormal levels of reduced thiols and TBARS was correlated with more severe disease during admission [33, 34].

Decreased plasma levels of reduced thiols and abnormalities of the thiol/disulphyde homeostasis were the most studied redox features in SARS-CoV-2 infection and represent a direct measure of protein oxidation [9, 35]. The assessment of TBARS is a non-specific but highly sensitive measure of lipid peroxidation that has been established as a global estimation of the sample oxidation level. Free radicals can attack polyunsaturated fatty acids (PUFAs) leading to the formation of lipid peroxides that can interact with other fatty acids, propagating the process [36, 37].

We found that the differences in reduced thiol levels for fatal and non-fatal events in the medium and long term were not as consistent as those during acute SARS-CoV-2 infection. However, TBARS levels were higher for fatal events both during admission and mid-term with results reaching statistical significance. Given the complexity of redox balance and the large number of involved factors, the differences observed between the two markers could have several readings.

The thiol pool is a major extracellular buffer for excess plasma ROS and represents the first line of defense to curb the impact of an acute redox imbalance [38]. The depletion of reduced thiols faced with a redox imbalance could be abrupt since the extracellular concentration of reduced thiols is substantially lower than the intracellular levels [9]. However, in the presence of reducing power, the glutathione pathway efficiently replenishes sulfhydryl groups at a rate that attempts to compensate for their consumption to reestablish the balance between the production of ROS and the reduced thiol pool [39].

An increased ROS production enhances the excessive formation of TBARS at a greater rate than the capacity of plasma antioxidant systems to neutralize them and this mismatch between production and clearance leads to their accumulation during acute processes. The natural evolution of TBARS levels consists of an initial increase in plasma concentrations that return to normal levels within weeks or months [40, 41]. However, there is a tendency toward persistently elevated plasma TBARS levels in some clinical processes and many of these clinical situations have in common the indefinite persistence of low-grade inflammation [42].

The results were quite unspecific as the number of factors that could enhance medium-long term abnormalities in oxidative stress markers increases progressively with the time after SARS-CoV-2 infection [43]. However, differences in the levels of some redox markers in the medium term would add to other clinical and analytical abnormalities extended over time that have been observed after SARS-CoV-2 infection. In this line, the presence of remnant and underlying inflammation in predisposed individuals could be behind syndromes such as the long or persistent COVID [44].

We observed differences in PTC, ferritin and LDH levels between some groups of patients, especially in the mid and long term outcomes. Although in some cases the results did not correspond to differences in TBARS or reduced thiols between the groups, these inflammatory markers might suggest the persistence of endothelial dysfunction, platelet activation and redox imbalance after SARS-CoV-2 infection as in other processes already explored [20, 45]. As time

since SARS-CoV-2 infection progresses, other variables became relevant in the prognostic differences between the groups. The greatest impact on morbidity and mortality in elderly individuals with multiple CVRFs and comorbidities is consistent with the presence of some chronic diseases with intercurrent exacerbations [46].

More controversial could be the correlation found between the time of adaptative immune response development and the levels of plasma reduced thiols during admission. A decrease in reduced thiols has already been involved in increased viral virulence due to a greater ability to enter the cell via the ACEI receptor [47]. The higher rate of SARS-CoV-2 entry into the cell due to and abnormal thiol/disulphite balance may be a determinant for an earlier and more intense immune response, which in turn has been associated with a severe inflammatory response and clinical pictures of worse prognosis, although further studies are needed [33].

Factors influencing the time to a negative RT-qPCR have been extensively studied and stronger inflammatory responses were associated with better viral clearance rates [24]. However, there is insufficient evidence to suspect that redox imbalance may be involved and we found no differences in SARS-CoV-2 clearance time between the comparison groups based on TBARS or reduced thiol levels.

The role of redox imbalance in the prognosis of SARS-CoV-2 infection in the short, medium and long term remains underexplored. The relationship between unfavorable redox status and the presence of persistent, underlying low-grade inflammation may partly explain some fatal and non-fatal events following SARS-CoV-2 infection as in other clinical processes. The role of SARS-CoV-2 infection in the inflammatory response has already been addressed but factors related to the adaptive immune response appear complex and less known. Redox imbalance might be one more variable in the efficiency of the specific immune response. However, more scientific evidence is needed.

## Limitations and strengths

This was a prospective single-center study of real clinical practice conducted in white elderly patients with SARS-CoV-2 infection from the northwest region of Spain (Galicia). Our results should be interpreted with caution when applying them to other population, race or ethnicity.

The selection of a small sample from the total number of patients admitted to the unit was a limitation that could influence the validity and accuracy of the results. Clinical and laboratory parameters, including redox markers, were measured at one point without monitoring. Although we attempted to identify the factors that could be involved in the redox status and prognosis of SARS-CoV-2 infection, some relevant variables may not have been explored. The existence of differences between the comparison groups could also be a limitation to the results of the survival analysis. Furthermore, some results should be viewed only as a clinical trend due to the limited number of events in some categories.

The assessment of TBARS and reduced thiols are only two of the multiple oxidative stress markers available to evaluate plasma oxidation and may therefore reflect a partial view of the true redox status. Furthermore, decreased reduced thiols and increased TBARS levels have low specificity for protein oxidation and lipid peroxidation, respectively. However, these procedures are quite sensible and offer a global estimation of plasma redox imbalance. The use of azithromycin and other acute treatments was higher in patients with a worse prognosis. However, A poor clinical evolution may justify a greater use of these therapies. The lower frequency of obesity in patients with worse outcomes coincided with a clinical profile of protein-calorie malnutrition in these patients on a case-by-case review which could partly explain the correlation with a worse prognosis.

## Acknowledgments

We would like to express our gratitude to the patients who participated in the study and their families for their support. We would like to thank the University Hospital and the University of Santiago de Compostela for the support provided.

## Author Contributions

**Conceptualization:** Nestor Vazquez-Agra, Anton Cruces-Sande, Alvaro Hermida-Ameijeiras.

**Data curation:** Nestor Vazquez-Agra, Ana-Teresa Marques-Afonso, Ignacio Novo-Veleiro.

**Formal analysis:** Nestor Vazquez-Agra, Ana-Teresa Marques-Afonso, Ignacio Novo-Veleiro.

**Investigation:** Nestor Vazquez-Agra, Anton Cruces-Sande.

**Methodology:** Nestor Vazquez-Agra, Anton Cruces-Sande.

**Supervision:** Antonio Pose-Reino, Estefania Mendez-Alvarez, Ramon Soto-Otero, Alvaro Hermida-Ameijeiras.

**Validation:** Antonio Pose-Reino, Estefania Mendez-Alvarez, Ramon Soto-Otero, Alvaro Hermida-Ameijeiras.

**Writing – original draft:** Nestor Vazquez-Agra.

**Writing – review & editing:** Nestor Vazquez-Agra.

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
