## [Decision Letter · Decision Letter 0]

8 Jun 2022

PONE-D-22-13272Assessment of oxidative stress markers in elderly patients with SARS-CoV-2 infection and potential prognostic implications. An observational studyPLOS ONE

Dear Dr. Agra,

Thank you for submitting your manuscript to PLOS ONE. After careful consideration, we feel that it has merit but does not fully meet PLOS ONE’s publication criteria as it currently stands. Therefore, we invite you to submit a revised version of the manuscript that addresses the points raised during the review process. One reviewer have serious concerns about writing style and the organization of different part of the  manuscript the other reviewer has concerns about the bimarkers used to asses the oxidative stress. Also i suggest to highlight the novelty of the manuscript very clearly as this has been addressed already in the literature. 

We look forward to receiving your revised manuscript.

Kind regards,

Gheyath K. Nasrallah

Academic Editor

PLOS ONE

Journal Requirements:

Reviewers' comments:

Reviewer's Responses to Questions

**Comments to the Author**

1. Is the manuscript technically sound, and do the data support the conclusions?

Reviewer #1: No

Reviewer #2: Partly

2. Has the statistical analysis been performed appropriately and rigorously? 

Reviewer #1: I Don't Know

Reviewer #2: I Don't Know

3. Have the authors made all data underlying the findings in their manuscript fully available?

Reviewer #1: Yes

Reviewer #2: Yes

4. Is the manuscript presented in an intelligible fashion and written in standard English?

Reviewer #1: No

Reviewer #2: Yes

5. Review Comments to the Author

Reviewer #1: In the manuscript entitled "Assessment of oxidative stress markers in elderly patients with SARS-CoV-2 infection and potential prognostic implications. An observational study", the authors presented an attempt to evaluate the possible correlation between several biochemical and physiological parameters and SARS-CoV-2 infection in elderly patients. I have major and minor concerns regarding the manuscript in its current form. Here are SOME of my major concerns:

1- The collected data is rich and numerous. However, the collected data definitely needs deeper analyses.

2. The interpretation of the data is not always clearly supported by the results.

3. Not all data is discussed and analyzed.

4. I am not an expert on statistics, but I feel that the statistical analysis provided in the manuscript needs double-checking given the complexity of the correlation among different tested parameters.

5. The organization of the manuscript, especially in the RESULTS section, is very weak. The sub-sections are literally "2 words" each!

6. The figures are very hard to read and analyze.

7. The writing of the manuscript needs major revision. The English language and the structuring of the sentences is very poor.

Reviewer #2: This is an interesting paper exploring a very intriguing correlation, ie the one between Oxidative Stress and COVID-19 infection

The only issue with this paper is that as compared to those used in many other papers, the biomarker of oxidative stress used here are not among the ones that are currently employed and are only two. Indeed, the majority of papers dealing with plasmatic biomarkers of oxidative stress use a number of different biomarkers e.g. 8-iso-prostaglandin F2α (8-iso-PGF2α), advanced oxidative protein products (AOPPs), protein carbonyl (PCO), glutathione peroxidase-3 (GPX-3), paraoxonase-1 (PON1). Other papers also analyze superoxide dismutase (SOD), Malondialdehyde (MDA), Glutathione reduced (GSH), and oxidated (GSSG).

In my opinion, this manuscript requires the assessment of, at least, another biomarker among those mentioned above in order to make a more solid correlation between Oxidative Stress and COVID-19 infection.

6. PLOS authors have the option to publish the peer review history of their article (what does this mean?). If published, this will include your full peer review and any attached files.

Reviewer #1: No

Reviewer #2: No

---

## [Author Response · Author response to Decision Letter 0]

25 Aug 2022

Dear Editor and Reviewers,

First, we would like to thank you for giving us the opportunity to share our research with you and your editorial group and for your time and dedication in the evaluation of the manuscript. Your recommendations, suggestions and advice were very much considered and undoubtedly contributed to improving the study. 

To the attention of the Editor:

Here is a detailed list of the responses to your concerns with the appropriate amendments.

We ensure that our manuscript met PLOS ONE's style requirements, including those for file naming, and followed the style templates.

We have provided the required details regarding participant consent in the ethics statement in the Methods section and online submission information. 

Section: Material and Methods. Subsection: Ethical approval: Paragraph Nº 1: “””. . .The Research Ethics Committee of Santiago de Compostela-Lugo approved this study (reference number 2020/578). All procedures were in accordance with the ethical standards of the responsible committee on human experimentation and with the Helsinki Declaration of 1975. Written informed consent was obtained from all patients for being included in the study. . .”””

The data was not a core part of the research being presented in the study, so we agreed to remove the phrase that referred to these data in the results section.

Section: Results. Paragraph Nº 1: “””. . . We included 61 patients (57% women) with a mean age of 83 years, and among them, 42 (69%), 31 (51%) and 15 (25%) had AHT, HLP and DM, respectively. A total of 27 (44%) suffered from chronic heart failure (HF). Approximately one of four individuals had tobacco or alcohol abuse. . .”””

4. If possible, can you please provide an email address and/or URL contact for the Galician (Spain) Health Service (SERGAS)/ Research Ethics Committee of Santiago de Compostela-Lugo (2020/578) where data access requests can be sent?

Data relevant to the results will be fully anonymized and incorporated into a repository recommended by your publisher.

At the end of the manuscript: 

“””…Data Availability

The anonymized database will be available from Open Science Framework (DOI: 10.17605/OSF.IO/R3S8U)...”””

To the attention of the Reviewer 1:

After carefully reading the article again and supported by your perspective, the authors agreed that the manuscript needed a profound restructuring of the statistical methodology and an inevitable redrafting of the results and discussion sections. Additionally, we have discussed the main limitations of the methodology and results in the corresponding subsection. Given the magnitude of the amendments, the response to all your concerns will be accompanied by representative text fragments of the complete modifications. Without further ado, we proceed to respond to your concerns.

1. The collected data is rich and numerous. However, the collected data definitely needs deeper analysis.

Given the unusual and worrying pandemic situation and the absence of scientific evidence in SARS-CoV-2 infection at that time, our objective was to carefully collect as much information as possible. We aimed to perform a descriptive analysis showing the most representative picture of the elderly patients admitted with SARS-CoV-2 infection to an Internal Medicine Department. However, we understand that total events at the 12th month was an ambiguous and non-specific variable that only yielded confusion, so we restructured the univariate and multivariate analysis by decomposing total events into fatal and non-fatal events during admission and at the 3rd 6th, and 12th months post-discharge.

Section: Material and Methods. Subsection: Outcomes. Paragraph Nº 1: """. . .Fatal events were referred to medical circumstances that led to the patient's death. We grouped them into the following categories: I) In-hospital fatal events; II) 3rd month post-discharge fatal events; III) 6th month post-discharge fatal events; and IV) 12th month post-discharge fatal events. Non-fatal events were grouped into the following categories: I) in-hospital non-fatal events, which referred to the need for transfer during admission to a critical respiratory care unit; and II) post-discharge non-fatal events, which referred to the need for readmission to hospital for respiratory or cardiovascular disease within the 3rd, 6th, or 12th month post-discharge. . ."""

We acknowledge that some results of the univariate analysis were not evaluated in depth since the total number of variables collected far exceeded the number of variables that it would be advisable to combine for a good performance in the multivariate analysis [1]. To increase the analysis dept, we decided to limit the number of included clinical and laboratory variables to those that could have the greatest explanatory capacity or influence on the results.

The small number of observations in some categories also limited the inclusion of some variables in the binary logistic regression for deeper analysis. As we explained in the text, it is possible that some variables evaluated in the univariate analysis were excluded from the multivariate analysis. However, we tried to ensure that those variables collected that could influence the redox balance and the outcomes were included.

Section: Discussion. Subsection: Limitations and strengths. Paragraph Nº 2: """...The selection of a small sample from the total number of patients admitted to the unit was a limitation that could influence the validity and accuracy of the results… Although we attempted to identify the factors that could be involved in the redox status and prognosis of SARS-CoV-2 infection, some relevant variables may not have been explored…"""

2. The interpretation of the data is not always clearly supported by the results.

We apologize if at any time we were too enthusiastic or categorical in the interpretation of the results. Given the limitations of the multivariate analysis, we understand that some comments in the discussion may not have been adequately supported by the results. With the redrafting of the manuscript, we intended to offer the results with greater resolution, avoiding the concept of total events at the 12th month and focusing on fatal and non-fatal events during admission and at the 3rd, 6th and 12th months post-discharge. The new organization of the results also entailed rebuilding of the discussion section to deal with each outcome in a more concrete and detailed way.

3. Not all data is discussed and analyzed.

We found a mismatch between the outcomes that were evaluated in the univariate and multivariate analysis so that fatal and non-fatal events during admission and at the 3rd, 6th, and 12th months post-discharge were not seen in depth, while the 12th month total events were not explicitly evaluated in the univariate analysis. As we explained in previous points, we avoided general outcomes and divided this variable into eight different variables (fatal and non-fatal events during admission and at the 3rd, 6th, and 12th months post discharge). Additionally, the results of the univariate analysis were examined in depth using a multivariate procedure. The discussion focused on the results that reached clinical relevance or statistical significance in the multivariate analysis.

4. I am not an expert on statistics, but I feel that the statistical analysis provided in the manuscript needs double-checking given the complexity of the correlation among different tested parameters.

Thanks to your comments, we realized that the statistical methodology subsection needed a thorough evaluation and the implementation of changes in both format and content, so we worked on a deep remodeling of the subsection. Two researchers conducted the statistical procedures independently, achieving concordance in the results. The medical records were reviewed to reduce missing values to below 5% and the remaining data were imputed as the sample median or mean depending on the distribution. After a new in-depth exploratory analysis of the quantitative variables, most of the analytical variables failed to meet some assumption of normality. Since parametric tests provide more powerful results than non-parametric tests, we first attempted to perform a logarithmic transformation of these non-normally distributed variables. The univariate results of the log-normally distributed variables were them back-transformed and expressed as mean and 95%CI [2]. The main methodological amendments are detailed as follows: 

A) Detailed description of the methodology related to the descriptive and univariate analysis.

Section: Material and Methods. Subsection: Statistical analysis. Paragraph nº 1: """. . . Statistical analysis was performed using SPSS 22.0 statistical software (SPSS Inc, Chicago, IL). First, we performed a descriptive analysis in which the frequencies of qualitative variables were expressed as number (n) and percentage (%). The Kolmogorov-Smirnov test was used to determine whether quantitative continuous variables were normally distributed. For those variables with a non-normal distribution, we performed a logarithmic transformation. Normally distributed variables were expressed as mean (m) and standard deviation (± SD), normally distributed variables after transformation were back-transformed and expressed as mean with 95% confidence interval (95%CI) and non-normally distributed ones were expressed as median and interquartile range (IQR). A missing value analysis was carried out on those variables with more than 5% missing values. We performed a comparative analysis between the groups of patients according to the outcome variables. The chi-square test was used to compare categorical variables, while quantitative variables were compared using the Student’s t-test or Mann–Whitney U test as appropriate. . ."""

B) Re-foundation of the multivariate analysis using binary logistic regression for fatal and non-fatal events during admission and at the 3rd, 6th and 12th months post-discharge and including in the models the variables that could influence redox markers and outcomes. We removed the results corresponding to total events, including the ROC curve model due to redundancy, unspecificity and ambiguity. 

Section: Material and Methods. Subsection: Statistical analysis. Paragraph no. 2: """. . . We performed a binary logistic regression analysis for fatal and non-fatal events during admission and at the 3rd, 6th and 12th months post-discharge using a non-automatic analysis procedure. Variables that showed clinical or statistical relevant differences (P-value < 0.1) in the univariate analysis were included. The validity of the model was evaluated using the Omnibus test in which a P-value of lower than 0.05 was considered necessary to assume that the current model was better than the null model. We provided a model summary with the deviance (-2 log-likelihood ratio test (-2LL)), coefficient of determination (R2) and the overall accuracy score. The parameters of those variables in the model were the non-standardized Beta coefficients (B), P-value of the Wald test and 95%CI for the coefficients. The variables with a P-value of lower than 0.1 (P< 0.150) were kept in the model. A P-value of lower than 0.05 (P< 0.05) was considered for statistical significance. . ."

C) Review of the survival analysis with emphasis on a better description of the statistical methodology and its application to our data. 

Section: Material and Methods. Subsection: Statistical analysis. Paragraph nº 3: """. . .We developed a survival analysis model using the Kaplan–Meier estimator for time to reach negative RT-qPCR and time to achieve significant anti-SARS-CoV-2 IgM titers during admission based on plasma levels of TBARS and reduced thiols and using as cut-off points the physiological limits of TBARS (4.0 µmol/L) and reduced thiol (0.40 mmol/L) levels. We compared the evolution of the variables of interest as a function of time between patients with lower or equal levels and higher levels of the oxidative stress markers. The relevance of the results was evaluated using the Log-Rank test and a P-value of lower than 0.05 (P< 0.05) was considered for statistical significance. . ."""

5. The organization of the manuscript, especially in the RESULTS section, is very weak. The sub-sections are literally "2 words" each!

The results section was redrafted on the basis of the profound modifications in the statistical methodology as follows:

A) As we provided in the text, the subsections corresponding to the univariate analysis for fatal and non-fatal events described in depth the differences between the comparison groups considering the methodological modifications.

Section: Results. Subsection: Fatal events. Paragraph No. 3: """. . . Patients who had a fatal event showed higher TBARS levels than controls measured in μmol/L and expressed as mean and 95%CI. These differences reached relevant results in the following groups: In-hospital (no: 2.84 (2.03–3.97), yes: 4.20 (2.93–6.01); P= 0.029), 3rd (no: 2.83 (2.00–3.98), yes: 3.69 (2.71–5.04); P= 0.054) and 6th (no: 2.81 (2.02–3.92), yes: 3.57 (2.42–5.26); P= 0.058) months post-discharge fatal events. The levels of reduced thiols measured in mmol/L and expressed as mean and 95%CI were lower in patients who suffered a fatal event. Such differences reached relevant results in the following groups: 3rd (no: 0.47 (0.40–0.56), yes: 0.42 (0.37–0.46); P= 0.059), 6th (no: 0.48 (0.40–0.56), yes: 0.41 (0.36–0.48); P= 0.026) and 12th(no: 0.48 (0.41–0.56), yes: 0.43 (0.36–0.51); P= 0.034) month post-discharge fatal events. The results are shown in Fig 1A and 1B.. . ."""

Section: Results. Subsection: Non-fatal events. Paragraph No. 3: """. . .Patients who had a non-fatal event presented higher TBARS levels than controls measured in μmol/L and expressed as mean and 95%CI. Such differences reached relevant results in the following groups: 3rd (no: 2.89 (2.03–4.13), yes: 3.31 (2.93–3.73); P= 0.074) and 6th (no: 2.77 (1.96–3.91), yes: 3.31 (2.37–4.62); P= 0.037) months post-discharge non-fatal events. The levels of reduced thiols measured in mmol/L and expressed as mean and 95%CI were lower in patients who suffered non-fatal events. Such differences reached relevant results in the following groups: In-hospital (no: 0.47 (0.40–0.56), yes: 0.37 (0.32–0.43); P= 0.019) non-fatal events. The results are extended in Fig 1C and 1D.…"""

B) In accordance with the methodological modifications, the tables for the results of the univariate analysis were completely redrawn. In order to reduce confusion in the presentation of the results, we eliminated the supplementary tables and completed the main tables with the most relevant clinical and analytical variables.

C) For the results of the multivariate analysis, we include a brief description of the binary logistic regression models focusing on the variables included, model reliability, summary statistics and parameters of the relevant variables. We also summarized the results in Table 3.

Section: Results. Subsection: Multivariate analysis. Paragraph 1: """...We performed a multivariate analysis for fatal and non-fatal events during admission and at the 3rd, 6th and 12th months post-discharge using binary logistic regression. We considered some clinical and laboratory variables that could influence TBARS and reduced thiol levels or be relevant to the outcomes as follows: age, sex, obesity, toxic habits (tobacco abuse and alcohol intake), CVR variables (AHT, DM and HLP), HF, Barthel index, chronic treatments (statins and RAAS blockers), acute treatments (Azithromycin and systemic steroids) and levels of some analytical parameters (FPG, creatinine, albumin, ferritin, fibrinogen and LDH). Table 3 shows the model summaries and the parameters of those variables that were kept in the model for fatal and non-fatal events during admission and at the 3rd, 6th and 12th months post-discharge…"""

D) For the survival analysis, we made a detailed description of the evolution of the variable studied over time based on the levels of the redox markers. 

Section: Results. Subsection: Correlation between oxidative stress markers and time to a negative RT-qPCR. Paragraph Nº 1: """… Fig 2 represents a survival analysis that shows on the ordinate axis the percentage of patients with a positive RT-qPCR for SARS-CoV-2 and on the abscissa axis the time to evolution in days since the hospital admission. At the beginning, the percentage of patients with a positive RT-qPCR for SARS-CoV-2 was 100% and progressively decreased with the evolution of the disease. The black curve in Fig 2A refers to patients with reduced thiol levels equal to or less than 0.40 mmol/L, while the gray curve refers to individuals with reduced thiol levels higher than 0.40 mmol/L. The black curve in Fig 2B refers to patients with TBARS levels higher than 4.0 µmol/L, while the gray curve refers to individuals with TBARS levels equal to or less than 4.0 µmol/L. The Log-Rank test showed that there were no differences in the median of days to achieving a negative RT-qPCR for SARS-CoV-2 based on different levels of TBARS or reduced thiols…"""

Section: Results. Subsection:Correlation between oxidative stress markers and time to significant anti-SARS-CoV-2 IgM titers. Paragraph Nº 1: """… Fig 3 represents a survival analysis that shows on the ordinate axis the percentage of patients with anti-SARS-CoV-2 IgM levels below the median (27 U/mL) and on the abscissa axis the time of evolution in days since the hospital admission. At the beginning, the percentage of patients with non-significant titers was 100% and progressively decreased with the evolution of the disease. The black curve in Fig 3A refers to patients with reduced thiol levels equal to or less than 0.40 mmol/L, while the gray curve refers to individuals with reduced thiol levels higher than 0.40 mmol/L. The black curves in Fig 3B refer to patients with TBARS levels higher than 4.0 umol/L, while the gray curves refer to individuals with TBARS levels equal to or less than 4.0 umol/L. We found that the median time to achieving anti-SARS-CoV-2 IgM titers of higher than 27 U/mL was 22 and 14 days for patients with reduced thiol levels greater than 0.40 mmol/L and lower than or equal to 0.40 mmol/L, respectively…"""

6. The figures are very hard to read and analyze.

We made changes in the format of the figures to improve readability and interpretation, including font modification, change to the required file format and dimension setting.

A) Fig 1: We increased the font size, removed the images corresponding to total events and corrected some mistakes in the labels since they did not coincide with the results shown in the text.

B) Fig 2: Based on the previous points, we decided to disregard the ROC curve model.

C) Fig 3 was renamed Fig 2. We incorporated a concise caption and a legend describing in detail the variables evaluated. The information in the figure is completed with that of the text in the corresponding subsection.

Section: Results. Subsection: Correlation between oxidative stress markers and time to a negative RT-qPCR. Fig 2: “””. . . % positive RT-qPCR– Percentage of patients with a positive Real-time reverse transcriptase-polymerase chain reaction for SARS-CoV-2. . .”””

D) Fig 4 was renamed Fig 3. We incorporated a concise caption and a legend describing in detail the variables evaluated. The information in the figure is completed with that of the text in the corresponding subsection.

Section: Results. Subsection: Correlation between oxidative stress markers and time to significant anti-SARS-CoV-2 IgM titers. Fig 3: “””...% SARS-CoV-2 IgM ≤ 27 U/mL–Percentage of patients with anti-SARS-CoV-2 IgM titers equal to or lower than 27 U/mL...”””

7. The writing of the manuscript needs major revision. The English language and the structuring of the sentences is very poor.

After your advice, we subscribed to a premium version of a prestigious next-gen grammar correction and language enhancement writing assistant designed for academic and technical writing named Trinka AI with the subscription ID sub_1LRZjGBOAkLRyixTEeCTbnhM (https://www.trinka.ai/es/). Many of the specific structural modifications implemented in the manuscript in the different sections corresponded to the recommendations of the writing assistant.

Closing remarks

Dear Reviewer 1.

We endeavored to adhere to your concerns in making all the relevant changes, which have meant a real revolution in the design of the manuscript. However, we firmly believe that the new version is better than the previous one and we thank you very much for your collaboration. We would only like to emphasize that the essence of the article and therefore an aspect that differentiates it from other studies on oxidative stress and prognosis in SARS-CoV-2 infection is the insight it provides in the medium and long term on a possible role of redox imbalance in the prognosis of patients beyond the acute stage of SARS-CoV-2 infection.

Thank you very much in the name of the group.

Kind regards;

Nestor Vazquez Agra

To the attention of the Reviewer 2:

Thank you for your contribution to this manuscript. Without further ado, we proceed to address your concerns.

1. The only issue with this paper is that as compared to those used in many other papers, the biomarker of oxidative stress used here are not among the ones that are currently employed and are only two. Indeed, the majority of papers dealing with plasmatic biomarkers of oxidative stress use a number of different biomarkers e.g. 8-iso-prostaglandin F2α (8-iso-PGF2α), advanced oxidative protein products (AOPPs), protein carbonyl (PCO), glutathione peroxidase-3 (GPX-3), paraoxonase-1 (PON1). Other papers also analyze superoxide dismutase (SOD), Malondialdehyde (MDA), Glutathione reduced (GSH), and oxidated (GSSG).

In my opinion, this manuscript requires the assessment of, at least, another biomarker among those mentioned above in order to make a more solid correlation between Oxidative Stress and COVID-19 infection.

When assessing oxidative stress, the quantification of reactive oxygen species (ROS) is very laborious and inaccurate due to their instability and short half-life. However, the reaction of free radicals with organic molecules generates organic products derived from the oxidation of carbohydrates, lipids, proteins and nucleic acids that are more stable and allow a more accurate assessment of oxidative stress [3].

Malondialdehyde (MDA), 8-iso-prostaglandin F2a (8-iso-PGF2a) and 4-hydroxynonenal (4-HNE) among others are by-products of lipid peroxidation and the assessment of TBARS is a measure of MDA levels. Although most MDA comes from lipid peroxidation of polyunsaturated fatty acids (PUFAs), it can also be the end product of oxidation of other biomolecules oxidation such as proteins. Therefore, as mentioned in the text, the weakness of these markers is their lack of specificity. However, it provides an estimation of the level of lipid peroxidation and an overall view of the level of oxidation of the sample [4].

The assessment of reduced (GSH) and oxidized (GSSG) Glutathione, thiol/disulphide balance and reduced thiols are estimators of the plasma pool of sulfhydryl groups, which are one of the main lines of defense against oxidative stress. Although the major source of thiols are plasma proteins and specifically albumin, there are free thiols and thiols forming part of other biomolecules such as glutathione or carbohydrates. The measurement of reduced plasma thiols in a non-proteinized sample is a measure of the level of protein oxidation. However, as we discussed for TBARS, the decrease in reduced thiols is the common pathway of several biomolecuolas oxidation. Thus, it provides an estimation of the level of protein oxidation and an overall view of the level of oxidation of the sample [5].

The evaluation of TBARS and reduced thiols during SARS-CoV-2 infection and their possible impact in the short, medium or even long term seemed very attractive to us as it had not yet been addressed and provided us with a global and very sensitive view of the levels of plasma oxidation in patients with SARS-CoV-2 infection. However, the measurement of other biomolecules derived from oxidative stress or the implementation of enzymatic method to evaluate the redox status would have provided greater value to the results and strength to the conclusions. Additionally, the use of more specific techniques, such as those based on chromatography, would have added specificity and precision to the estimates. 

The limited number of techniques and their lack of specificity were limitations that we noted in the corresponding section.

Section: Discussion. Subsection: Limitations and Strengths. Paragraph Nº 3: """. . .The assessment of TBARS and reduced thiols are only two of the multiple oxidative stress markers available to assess plasma oxidation and may therefore reflect a partial view of the true redox status. Furthermore, decreased reduced thiols and increased TBARS levels have low specificity for protein oxidation and lipid peroxidation, respectively. However, these procedures are quite sensible and offer a global estimation of plasma redox imbalance. . ."""

Closing remarks

Dear Reviewer 2.

As you well commented, most studies on oxidative stress and SARS-CoV-2 employ a greater number and variety of redox markers, thus providing more robust results on short-term prognosis during SARS-CoV-2 infection. However, our main objective was to globally evaluate the impact of a prooxidant internal milieu during SARS-CoV-2 infection on prognosis also in the medium and long term. We are aware of these limitations and it is our duty to highlight them.

Thank you very much in the name of the group.

Kind regards;

Nestor Vazquez Agra

References

1. Stoltzfus JC. Logistic regression: a brief primer. Acad Emerg Med. 2011;18: 1099–1104. doi:10.1111/j.1553-2712.2011.01185.x

2. West RM. Best practice in statistics: The use of log transformation. Ann Clin Biochem. 2022;59: 162–165. doi:10.1177/00045632211050531

3. Jakubczyk K, Dec K, Kałduńska J, Kawczuga D, Kochman J, Janda K. Reactive oxygen species - sources, functions, oxidative damage. Pol Merkur Lekarski. 2020;48: 124–127.

4. Tsikas D. Assessment of lipid peroxidation by measuring malondialdehyde (MDA) and relatives in biological samples: Analytical and biological challenges. Anal Biochem. 2017;524: 13–30. doi:10.1016/j.ab.2016.10.021

5. Turell L, Radi R, Alvarez B. The thiol pool in human plasma: the central contribution of albumin to redox processes. Free Radic Biol Med. 2013;65: 244–253. doi:10.1016/j.freeradbiomed.2013.05.050

---

## [Decision Letter · Decision Letter 1]

19 Sep 2022

Assessment of oxidative stress markers in elderly patients with SARS-CoV-2 infection and potential prognostic implications in the medium and long term

PONE-D-22-13272R1

Dear Dr. Agra, 

We’re pleased to inform you that your manuscript has been judged scientifically suitable for publication and will be formally accepted for publication once it meets all outstanding technical requirements.

Kind regards,

Gheyath K. Nasrallah

Academic Editor

PLOS ONE

Additional Editor Comments (optional):

Reviewers' comments:

Reviewer's Responses to Questions

**Comments to the Author**

1. If the authors have adequately addressed your comments raised in a previous round of review and you feel that this manuscript is now acceptable for publication, you may indicate that here to bypass the “Comments to the Author” section, enter your conflict of interest statement in the “Confidential to Editor” section, and submit your "Accept" recommendation.

Reviewer #1: All comments have been addressed

Reviewer #2: All comments have been addressed

2. Is the manuscript technically sound, and do the data support the conclusions?

Reviewer #1: Yes

Reviewer #2: Yes

3. Has the statistical analysis been performed appropriately and rigorously? 

Reviewer #1: I Don't Know

Reviewer #2: I Don't Know

4. Have the authors made all data underlying the findings in their manuscript fully available?

Reviewer #1: Yes

Reviewer #2: Yes

5. Is the manuscript presented in an intelligible fashion and written in standard English?

Reviewer #1: Yes

Reviewer #2: Yes

6. Review Comments to the Author

Reviewer #1: All my comments have been addressed. The revised manuscript is a much enhanced version compared to the original one. However, I must stress again that I am not an expert on statistics, and I cannot make any judgement on the accuracy of the statistical analyses provided in this manuscript. In addition, I still believe that the manuscript should be re-checked for any linguistic (typographical and grammatical) mistakes. For example, "adaptative immune response" should read "adaptive immune response", "However, A poor clinical evolution" should read "However, a poor clinical evolution", "due to and abnormal thiol/disulphite balance" should read "due to an abnormal thiol/disulphite balance", ...., etc.

Reviewer #2: The authors have addressed all my concerns. Thanks !...............................................

7. PLOS authors have the option to publish the peer review history of their article (what does this mean?). If published, this will include your full peer review and any attached files.

Reviewer #1: **Yes: **Amin F. Majdalawieh

Reviewer #2: No

---

## [Editor Report · Acceptance letter]

27 Sep 2022

PONE-D-22-13272R1 

Assessment of oxidative stress markers in elderly patients with SARS-CoV-2 infection and potential prognostic implications in the medium and long term 

Dear Dr. Vazquez-Agra:

I'm pleased to inform you that your manuscript has been deemed suitable for publication in PLOS ONE. Congratulations! Your manuscript is now with our production department. 

Kind regards, 

on behalf of

Dr. Gheyath K. Nasrallah 

Academic Editor

PLOS ONE